

# Review article: Potential of Nature-Based Solutions to Mitigate Hydro-Meteorological Risks in Sub-Saharan Africa

Kirk B. Enu[1], Aude Zingraff-Hamed[1], Mohammad A. Rahman[1], Lindsay C. Stringer[2], Stephan Pauleit[1]

[1]Chair for Strategic Landscape Planning and Management, Technical University of Munich, Germany
[2]Department of Environment and Geography, University of York, York, United Kingdom

*Correspondence to*: Kirk B. Enu (kirk.enu@tum.de)

**Abstract.** Sub-Saharan Africa (SSA) is the region most vulnerable to climate change and related hydro-meteorological risks. These risks are exacerbated in rapidly expanding urban areas due to the loss and degradation of green and blue spaces with their regulating ecosystem services. The potential of nature-based solutions (NBS) to mitigate hydro-meteorological risks such as floods is increasingly recognized in Europe. However, its application in urban areas of SSA still needs to be systematically explored to inform and promote its uptake in this region. We conducted a multidisciplinary systematic review following the PRISMA (Preferred Reporting Items for Systematic Reviews and Meta-Analyses) protocol to establish the general patterns in the literature on NBS and hydro-meteorological risk mitigation in SSA. We searched scientific journal databases, websites of 12 key institutions and 11 NBS databases and identified 45 papers for analysis. We found at least one reported NBS in 71 % of urban areas of SSA across 83 locations. 62% of the papers were clustered in South Africa, Kenya, Tanzania and Nigeria only, while the most studied cities were Dar es Salaam and Kampala. Moreover, 66 NBS practices were identified, most of which (n=44) were for flood mitigation. With only Mozambique (n=2) reporting NBS among the most at-risk countries, we found that NBS are implemented where risks occur but not where they are most severe. Mangrove restoration and wetland restoration, reforestation and urban forests, and agroforestry and conservation agriculture were the most common NBS practices identified for floods, extreme heat and drought mitigation, respectively. Traditional practices that fit the definition of NBS, such as grass strips and stone bunds, and practices more popular in the Global North, such as green roofs and green façades, were also identified. These NBS also provided ecosystem services, including 15 regulatory, 5 provisioning and 4 cultural ecosystem services, while 4 out of every 5 NBS created livelihood opportunities. We conclude that reported uptake of NBS for hydro-meteorological risks in SSA is low. However, there could be more NBS, especially at the local level, that are unreported. Furthermore, NBS can help SSA address major development challenges such as water and food insecurity and unemployment and help the sub-region develop climate-resiliently. We, therefore, recommend that NBS be mainstreamed into urban planning and for knowledge exchange opportunities between SSA and Europe and other regions to be explored to promote uptake.

**Keywords.** ecosystem services, floods, extreme heat, drought, green infrastructure, ecosystem-based adaptation, natural hazards



## 1. Introduction

Climate change, uncontrolled urbanization and associated biodiversity loss are among the most significant socio-ecological challenges confronting Sub-Saharan Africa (SSA) in the 21st Century. These challenges increase vulnerability to hydro-metrological hazards such as floods, storms, heatwaves, droughts and wildfires, which pose a significant hydro-meteorological risk (Malgwi et al., 2020). Hydro-meteorological risk refers to the probability of damage resulting from hydro-meteorological hazards based on the exposure and vulnerability of populations and the environment. Hydro-meteorological risks have become more pronounced in SSA in recent decades, and their impacts are already being felt across all sectors (Arias et al., 2021).

The Intergovernmental Panel on Climate Change (IPCC) has made many observations on Africa's climate (Gutiérrez et al., 2021). They report that North and Southern Africa could warm by 4°C or more and record a reduction in precipitation between 10-20% by 2080. Thus, both areas are most susceptible to extreme heat and drought events. East and Central Africa are expected to experience an increase in rainfall by 15% or more by 2080, thereby, being most susceptible to floods. The Sahel and the rest of SSA are expected to record a general increase in temperatures as well as precipitation. From 2000-2019, flooding incidences claimed thousands of lives, injured even more and destroyed properties worth millions and account for 64% of hazard events in SSA (Malgwi et al., 2021). Droughts have also impacted over 269.6 million people and accounted for 46% of climate-induced deaths, while heatwaves have equally affected many over the same period (CRED, 2019). These realities underscore the pressing need for swift climate action among the 48 SSA countries (World Bank, 2022).

Conventional engineering approaches like the construction of dikes and large drains for addressing flood hazards have long been favoured by decision-makers (Lucas, 2020). However, many researchers and practitioners agree that conventional engineering responses to floods and other hydro-meteorological risks produce sub-par outcomes (Depietri & McPhearson, 2017). Conventional engineering solutions are often effective only in the short term (Lafortezza et al., 2018; Zhongming et al., 2014). This is evidenced in the many reported cases of levees being overtopped by waves or completely failing due to internal erosion or instability not long after construction (Özer et al., 2016). Conventional engineering solutions are also comparatively capital-intensive and often negatively impact natural ecosystems. Coupled with increasing levels of environmental degradation and recognition of the need for more joined-up approaches that link climate change adaptation, mitigation and development (IPCC, 2022), nature-based solutions (NBS) are increasingly being considered as alternatives or complements to conventional engineering for mitigating risks (Deng et al., 2022; Kalantari et al., 2018; Lupp, Zingraff-Hamed, et al., 2021).

The European Commission has defined NBS as "actions inspired by, supported by, or copied from nature" (European Commission & Directorate-General for Research and Innovation, 2015 p. 5). Such actions can be implemented as site-specific interventions at local scales or transcend national, regional or even international boundaries in rural or urban areas (Lindley et al., 2018). Ultimately, the overarching objective of NBS is to address socio-ecological challenges, including climate change





and associated hydro-meteorological risks, food and water insecurity and health concerns, while helping locales to attain their
sustainable development aspirations.
In terms of operationalization, especially through the European Union Horizon 2020 (EU-H2020) programme, the application
of NBS in Europe has focused significantly on the restoration of degraded or lost ecosystems (EC, 2016), development of
green spaces and their socio-economic benefits (Matsler et al., 2021) and implementing solutions to hydro-meteorological
risks that mimic natural processes (Solheim et al., 2021). In SSA, conservation initiatives such as protecting green and blue
spaces have been considered to fall under the NBS umbrella (Thorn et al., 2021). This is appreciated in the more recent
definition of NBS by the 5th Session of the United Nations Environment Assembly as "actions to **protect**, **conserve**, restore,
sustainably use and manage natural or modified terrestrial, freshwater, coastal and marine ecosystems, which address social,
economic and environmental challenges effectively and adaptively, while simultaneously providing human wellbeing,
ecosystem services and resilience and biodiversity benefits" (Seddon, 2022). As of 2018, SSA was only 0.16% built-up
(Karamage et al., 2018) compared to 4.2% in Europe (EUROSTAT, 2021); thus, it is plausible that greater attention will be
on ecosystem conservation in SSA. Even though there are many definitions of NBS, its principles provide a common
understanding and framework for its implementation. NBS, therefore, particularly in urban settings, has to adopt a systems
approach (Stringer et al., 2018); mirror natural processes; produce multiple benefits for both people and biodiversity
(Somarakis et al., 2019); be inclusively designed, planned, implemented and managed; designed to fit the specific local context
in which it is applied; and support mutual learning for sustainability transitions (Kabisch et al., 2022).
In terms of the typologies of NBS, different approaches have been proposed. There are classifications by the level and type of
engineering applied, how biodiversity and ecosystems are managed, the stakeholders involved (Eggermont et al., 2015), or the
number of ecosystem services delivered (European Commission & Directorate-General for Research and Innovation, 2015).
NBS is also classified based on the problem it is deployed to solve, often in relation to the Sustainable Development Goals
(SDGs) (Somarakis et al., 2019). This study, however, adopts the classification by the kind of ecosystem the NBS is based in,
whether terrestrial or aquatic. On that account, there are green NBS which are vegetation-based, blue NBS, which are water-
based and hybrid NBS, which combine green and blue NBS within constructed (grey) structures (Sowińska-Świerkosz &
Garcia, 2022). This study also makes reference to NBS practices, which are conceived as activities, including those related to
planning, designing, implementation and management, that lead to the actual application of a NBS type. Such practices may
include river restoration efforts, rain gardens, green façades and permeable pavements (Zingraff-Hamed et al., 2020).
Many authors have demonstrated the effectiveness of NBS in urban areas. For instance, the effectiveness of NBS in slowing
runoff and reducing flood risk has been proven in Europe, North America (Pugliese et al., 2022) and Asia (Li & Zhang, 2022).
NBS has also shown effectiveness in reversing the effect of urban heat islands (Rahman et al., 2019), reducing erosion by up
to 90% (Keesstra et al., 2018), as well as improving air quality (Kim & Song, 2019).





In SSA, the fastest urbanizing region in the world (Moriconi-Ebrard et al., 2020), NBS offers the potential for mitigating hydro-
meteorological risks for several reasons. First, they tend to be cost-effective and are more efficacious over the long term. In
comparison to conventional engineering solutions, NBS can achieve up to 85% of profitable hydro-meteorological risks
management (Debele et al., 2019) and, in a broader context, could provide about 30% of the cost-effective mitigation required
to keep global warming below 2°C by 2030 (Seddon et al., 2019). This cost-effectiveness is vital for SSA, a region which's
climate adaptation efforts have been constrained by financial challenges (Gilder & Rumble, 2020). Second, NBS can deliver
multiple ecosystem services, ranging from provisioning to regulatory and cultural services (Pauleit et al., 2017). In contrast,
conventional engineering solutions often serve just one purpose, like wastewater treatment. Provisioning services are essential
given the high poverty levels in SSA and low employment rates, which mean that there is a high direct reliance on water, food
and energy. Third, leveraging NBS could help SSA to achieve the SDGs, particularly goals 11 (sustainable cities and
communities), 13 (climate action) and 15 (life on land). Fourth, NBS are important for SSA because the sub-region is home to
important biodiversity, some located in urban areas. Presently, over 33 major developments are proposed or under development
in different locations in SSA, including in major cities, which traverse 400 protected areas (Enns et al., 2019). Thus, embracing
NBS may hold the best prospects for addressing hydro-meteorological risks in SSA without compromising the natural system's
ability to support life (Archer et al., 2018).
Despite these potential benefits of NBS, it is unclear to what extent they have been implemented in SSA, including what NBS
types and specific practices have been used and for achieving what aims, especially in the context of increasing incidences and
severity of hazards. In the Global North, literature on NBS for hydro-meteorological risk mitigation is widespread through, for
instance, EU-H2020 projects like PHUSICOS, proGIreg, URBINAT, BiodivERsA, CleanUP and CleverCities (Ruangpan et
al., 2020; Schröter et al., 2021). However, literature on NBS in SSA is limited. Emerging studies focus mainly on incorporating
the concept into urban planning. Such studies are centred chiefly in South Africa (e.g., Molla, 2015; Russo et al., 2017; Venter
et al., 2020), leaving the rest of the sub-region, including some of the most at-risk countries, understudied. Furthermore, recent
systematic review studies were published on related concepts like green infrastructure and ecosystem services (Choi et al.,
2021; Douglas, 2018; Du Toit et al., 2018). There is a gap, therefore, in understanding how NBS can be applied for hydro-
meteorological risk mitigation in urban areas of SSA. This gap can be a significant setback to the uptake of the concept, which
is plausible in many ways for responding to hydro-meteorological risks and obtaining co-benefits. We conducted a systematic
review, therefore, to answer the following questions:





1. What is the extent of reported NBS uptake for hydro-meteorological risks mitigation in urban areas of SSA?
2. Are reported NBS being implemented where risks are located?
3. What specific NBS (types and practices) reported in the literature are being used to address floods, extreme heat and drought?
4. What other benefits are reported to accrue from these NBS beyond hazard risk mitigation through ecosystem services provision and livelihood generation?

## 2. Methods

### 2.1. Selection of papers

The research methodology consisted of several steps (Fig. 1). First, we identified peer-reviewed scientific articles satisfying the search criteria. Second, we accessed grey literature by searching websites of key institutions and NBS databases for NBS projects and initiatives to ensure that NBS advanced by development agencies but not scientifically studied were not missed. The peer-reviewed scientific articles were accessed through Scopus, Science Direct, Web of Science and Google Scholar. Grey literature was searched on the websites of 12 key institutions, including UN agencies and Local Governments for Sustainability (ICLEI), and 11 NBS databases (Supplementary Material Table 1). Following this paper selection process, eligibility was checked according to the inclusion and exclusion criteria, and a thematic analysis was carried out.



**Criteria for search process**
1- Papers on NBS for hydro-meteorological risk mitigation in Sub-Saharan Africa.
2- Peer-reviewed papers published from January 1, 2008 to December 31, 2021.
3- Grey literature published from January 1, 2008 to April 30, 2022

| Sourcing papers from Scopus, ScienceDirect, Web of Science & Google Scholar (n= 3,530) | Sourcing papers from key institutions (n=71) | Sourcing papers from NBS databases (n=688) |
|---|---|---|

Combining peer-reviewed papers and grey literature
(n= 4,289)

Removal of duplicates
(n=688)

Preliminary analysis
(n=3,601)

**Inclusion/exclusion criteria**

| Criteria | Included | Excluded |
|---|---|---|
| Publication type | Peer-reviewed papers, technical reports, book, policy brief, factsheet | Blog posts, news, magazine articles, commentaries, editorials, toolkits and training materials |
| Geography | Sub-Saharan Africa (according to World Bank's definition—48 countries) | Countries outside Sub-Saharan Africa—Algeria, Egypt, Libya, Morocco and Tunisia |
| Language | English, French | Any other language (e.g., Portuguese, Kiswahili, Arabic, etc.) |
| Hydro-meteorological risk addressed | Flood, extreme heat and drought | Storm surges, landslides, avalanches, hail, windstorms and forest fires |
| NBS typology (according to (Donatti et al., 2020) | On-the-ground actions (with or without enabling activities) | Only enabling activities |
| Text | Full text available | Only abstract available |

Screening of papers based on titles and keywords
(n=3,601)

Articles excluded after screening
(n= 3,312)

Records selected for abstract screening
(n = 289)

Articles excluded after screening
(n = 224)

Records selected for full-text review
(n = 65)

Articles excluded after full-text review

45 papers included in this review

**Identify trends, locations of NBS, specific NBS types and practices in use and benefits derived**



**Figure 1. Flowchart of literature screening and selection process.**
Search terms were informed by an initial scoping of the paper repositories and a review of NBS and green infrastructure
definitions, typologies and practices (Koc et al., 2017; Somarakis et al., 2019). Specific terms used during the search process
were related to nature-based solutions, green infrastructure, ecosystem services, urbanization, hydro-meteorological risks and
Sub-Saharan Africa (Table 1).
**Table 1. Terms used in different combinations for the literature search**

| Keyword | Related search terms |
|---|---|
| Nature-based solutions | Nature-based solutions, natural infrastructure, river protection, river conservation, river restoration, river management, flood management, flood mitigation, wetland conservation, wetland restoration, permeable pavement, permeable paving, infiltration basins, infiltration trenches, green roofs, rain garden, blue roof, urban wetland, French drain, low impact infrastructure, bio-retention, dry well, urban waterway, rain barrels and cisterns |
| Green infrastructure | Green infrastructure, green space, green spaces, low impact development, green infrastructure types, green streets, greenscape, naturalized landscaping, trees, urban forest, urban greening, urban parks |
| Ecosystem services | Ecosystem services, ecosystem protection, ecosystem conservation, ecosystem restoration, ecosystem management, ecosystem-based adaptation |
| Urbanization | Urbanization, urban growth, urban planning, spatial planning, land-use change |
| Hydro-meteorological risks | Climate change, climatic extremes, hydro-climatic extremes, hydro-meteorological risks, climate impacts, extreme events, extreme heat, extreme rainfall, heat mitigation, cooling, rainwater runoff, stormwater, surface runoff |
| Sub-Saharan Africa | sub-Saharan Africa |

*NB: Supplementary Material Table 1 contains the specific terms used for each database search.*
According to Donatti et al. (2020), NBS can be advanced as on-the-ground actions or enabling activities. On-the-ground actions
include ecosystem protection and restoration efforts, agricultural forest and conservation management practices; urban
gardens; and green infrastructures. Enabling activities focus on formulating policies, developing strategic plans, and
awareness-raising campaigns. In many cases, both approaches are married in NBS roll-out. However, the literature search
excluded papers that focused on only enabling activities since the study sought to document specific and tangible actions
implemented to help address hydro-meteorological risks.
The grey literature search was conducted on the websites of key institutions and the NBS databases from April 23-30, 2022.
Peer-reviewed scientific papers were searched using the Publish or Perish software, version 8.2, considering the time window
from January 1, 2008- December 31, 2021. These years were selected as 2008 was when the concept of NBS emerged



(Ruangpan et al., 2020). The literature search also allowed papers published in English and French, the top two official
languages used by countries in SSA. In all, 3,530 scientific peer-reviewed papers and 759 papers of grey literature were found.
## 2.2. Screening and eligibility selection
The screening was done by examining the titles and abstracts and, subsequently, the full text of the papers. The screening and
selection process followed the Preferred Reporting Items for Systematic Reviews and Meta-Analyses guidelines, according to
Page et al. (2021). Eligible papers had to meet the criteria defined in Fig. 1. Generally, papers included in the review had to
provide data on NBS that address specific hydro-meteorological risks; and have an SSA city or peri-urban area—as several
SSA countries lack a clear delineation of urban and rural areas—as study area (Du Toit et al., 2018).
Apart from project documents, technical reports, factsheets and policy briefs, non-peer-reviewed literature such as blog posts,
news, magazine articles, commentaries and editorials were excluded to ensure that only papers following scientific standards
were used for the review. Two people did the screening: one of the authors and a research assistant. Forty-five papers were
deemed eligible for the study. Out of them, 18 were peer-reviewed papers, while 27 were publications of grey literature. Only
one paper, a publication of grey literature, was published in French. The remaining papers were published in English.
## 2.3. Quality appraisal
The quality and strength of evidence are essential to the systematic review process (Movsisyan et al., 2018). Therefore, this
study used a 14-point framework to assess the quality of included papers (Supplementary Material Table 2). The framework
asked a series of questions on three themes—quality of reporting (six questions), risk of bias minimization (five questions)
and appropriateness of conclusions (three questions)—to ensure that quality research is done (Venkataramanan et al., 2018).
For each paper, a score of 0, 0.5 or 1 was given for each of the 14 questions, and the scores were then converted to percentages
to compare across themes (Supplementary Material Fig. 1). The studies were rated from the perspective of social-ecological
research methods as high quality (score of $\geq 10$ to 14), medium quality (score $\geq 5$ and $< 10$) or low quality (score $<5$).
## 2.4. Data extraction, presentation and analysis
The data from the selected papers were extracted into Notion version 2.0.21, a project management software developed by
Notion Labs Incorporated, for assessment. The coded information included:
- study title;
- author(s);
- year of publication;
- city/location;
- country;



• hydro-meteorological risks addressed;
• NBS practices and types used;
• ecosystem services(regulatory, provisioning and cultural) provided and
• livelihood generation (which was added later as an economic benefit of NBS after it was found to be a highly reported
variable across the papers).
A narrative summary of the papers is then given with the aid of tables, graphs and figures. ArcGIS Pro (version 2.8) by Esri
(2022) was used to create maps to visualize the location of NBS.
### 2.5.    Study limitation
By conducting this study using a systematic review methodology, we could establish general trends in the literature on NBS
and hydro-meteorological risks mitigation in urban areas of SSA. However, factors such as the finite selection of keywords
and poorly written abstracts could have led to the exclusion of important papers from the review. The study did not assess the
impacts of implemented NBS to determine whether they were successful or if any lessons can be drawn due to the lack of the
requisite data. In addition, the search was limited to only floods, extreme heat and drought, the most frequent hydro-
meteorological risks in SSA, although other risks like landslides and wildfires are recorded in the sub-region. Even though
excluded languages like Portuguese and Kiswahili are not as widely spoken as English and French in SSA, the exclusion of
papers published in these languages may also be a limitation of this study. Furthermore, because the study focused only on
reported NBS, it is likely that some implemented or ongoing NBS which are unreported were not captured.





## 3.   Results
### 3.1.   Extent of reported NBS for hydro-meteorological risk mitigation in SSA
#### 3.1.1.   Locations of papers

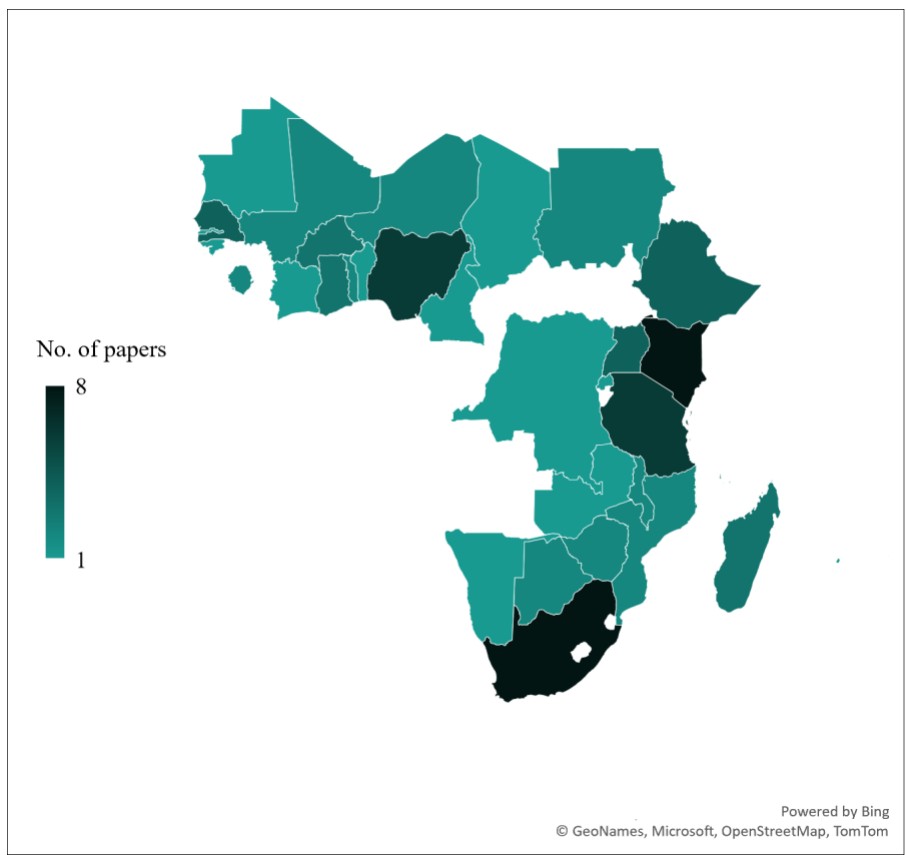

**Figure 2. Locations of papers on NBS for hydro-meteorological risks mitigation in SSA**
From 45 papers, the study found NBS used for hydro-meteorological risks mitigation in 34 SSA countries across 83 locations.
Thus, there is at least one reported NBS in 70.8% of urban areas of SSA countries. In terms of sub-regional distribution, 34.1%
of the papers (n=30) were from West Africa, 20.5% (n=18) from Southern Africa, 34.1% (n=30) from East Africa and 6.8%
(n=6) from Central Africa. Four papers (4.5%) were SSA-wide.
Countries with the most papers (62.2%) reporting NBS were South Africa (n=8), Kenya (n=8), Tanzania (n=6) and Nigeria
(n=6). The remaining countries had four or fewer papers, with 12 countries (35.3%) having only one paper. Cities with the
most reported NBS were Dar es Salaam (n=6) in Tanzania and Kampala (n=3) in Uganda. Nine cities (12.5%), including
Accra, Johannesburg and Nairobi, had two papers, while the remaining 63 locations (84.7%) had only one paper reporting on
them. Figure 2 gives a graphical representation of the locations of the papers.





### 3.1.2.    Risks addressed

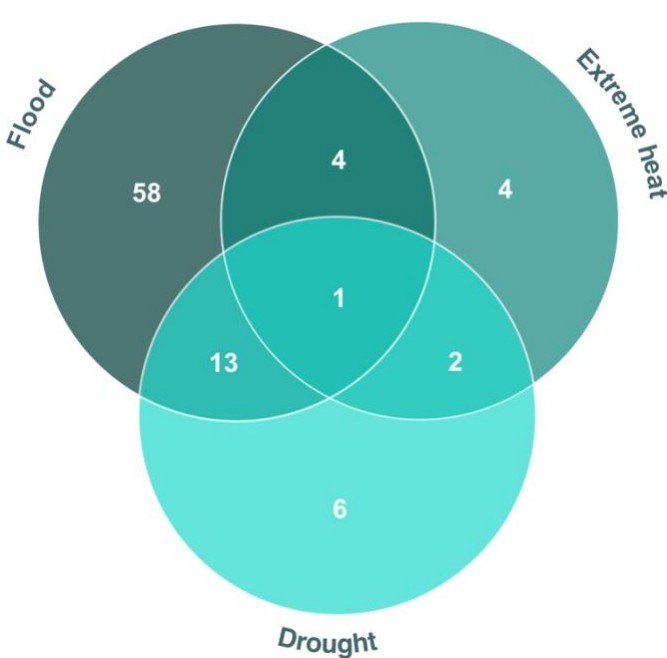

**Figure 3. Hydro-meteorological risks addressed with different NBS practices in SSA.**
A substantial number of the reported NBS (n=20) were intended to address more than one hydro-meteorological risk in their
implemented locations (Fig. 3). For instance, the marine conservation initiative in Johannesburg was found to address all three
risks studied (Washbourne, 2022). In Lagos, Nigeria, green conservation efforts were used to mitigate floods and extreme heat
(Mauvais, 2018). In cities like Dar es Salaam in Tanzania and Windhoek in Namibia, urban agriculture was used to address
floods and droughts (Thorn et al., 2021). Similarly, rainwater harvesting techniques across many countries, including Mali,
Chad, Sudan and Senegal, were used for flood and drought mitigation (Tamagnone et al., 2020).
### 3.1.3.    Scale of implementation
NBS in SSA were implemented over local (n=14), national (n=20), regional (n=3) and international scales (n=2) as indicated
in Fig. 4. Some papers did not specify the implementation scale of the reported NBS (n=6) for diverse reasons, including that
they were systematic reviews (e.g., Adegun et al., 2021; Choi et al., 2021) or conceptual papers (e.g., Kalantari et al., 2018).





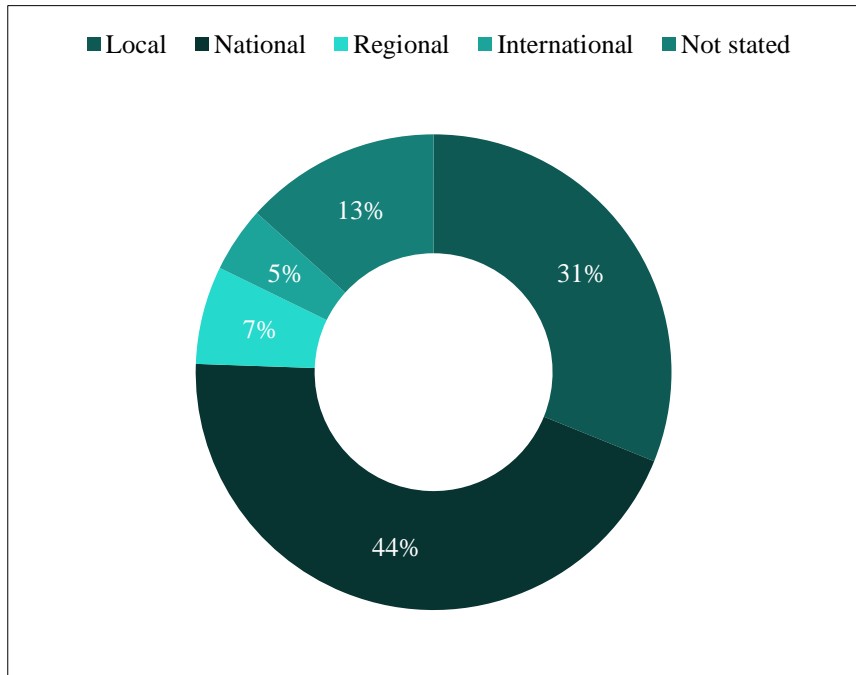

**Figure 4. Implementation scale of NBS.** Local-scale NBS are conceived as those implemented in specific local communities in a country, often by local actors, including non-profits (NGOs), community-based organizations (CBOs), local government administrations or the community. National NBS are implemented in different locations within the same country and are often advanced or coordinated by national agencies. Regional NBS refer to those that transcend two or more SSA countries. Lastly, international scale NBS are conceived as those implemented in SSA and countries on other continents.

Identified local NBS include reforestation and organic farming efforts in Obudu, Nigeria, used for addressing droughts and floods (UNDP, 2017) and several rainwater harvesting technologies used by communities in Burkina Faso, Chad, Mali, Mauritania, Niger, Senegal and Sudan, where drought and flash floods are major concerns (Tamagnone et al., 2020). Other examples are in Accra (Ghana), Dar es Salaam (Tanzania) and Kampala (Uganda), where urban agriculture was used to slow runoff and address flooding (Lwasa et al., 2014).

Local Action for Biodiversity is an example of a national NBS (ICLEI, 2010). This project was implemented in many locations across South Africa, including Cape Town, Durban and Cape Winelands and involved wetland conservation and restoration. The use of natural retention ponds and wetland conservation in Dakar, Senegal, to address floods advanced by the World Bank is also an example of a national NBS (Jongman et al., 2019).

Regarding regional NBS, the Great Green Wall is a good example (Turner et al., 2021). The project cuts across the entire width of Africa and spans 8,000 km of drylands in Burkina Faso, Chad, Djibouti, Eritrea, Ethiopia, Mali, Mauritania, Niger, Nigeria, Senegal and Sudan. The project seeks to rehabilitate lands through multifaceted afforestation, reforestation and revegetation measures, and sustainable agriculture. It is also expected to help mitigate climate change and address extremes such as drought



and extreme heat. Another example is the Urban Natural Assets for Africa by ICLEI, which used practices like mangrove
restoration, river restoration and green conservation to mitigate floods in locations across Tanzania, Mozambique, Uganda,
Malawi, Kenya and Ethiopia.
Two international scale NBS were identified. One is the Gazi Mangrove Restoration Project, implemented in Kenya and
Bangladesh to mitigate floods through mangrove restoration (Taylor & Oluoch, 2012). The other is the Ecosystem-Based
Adaptation in Marine, Terrestrial and Coastal Regions project, implemented in South Africa, Brazil and the Philippines
(CIFOR, 2013), exploring the effectiveness of wetland restoration, rangeland rehabilitation and the restoration of degraded
lands for flood mitigation.




1    **3.2.    Relationship between location of NBS and location of risks**

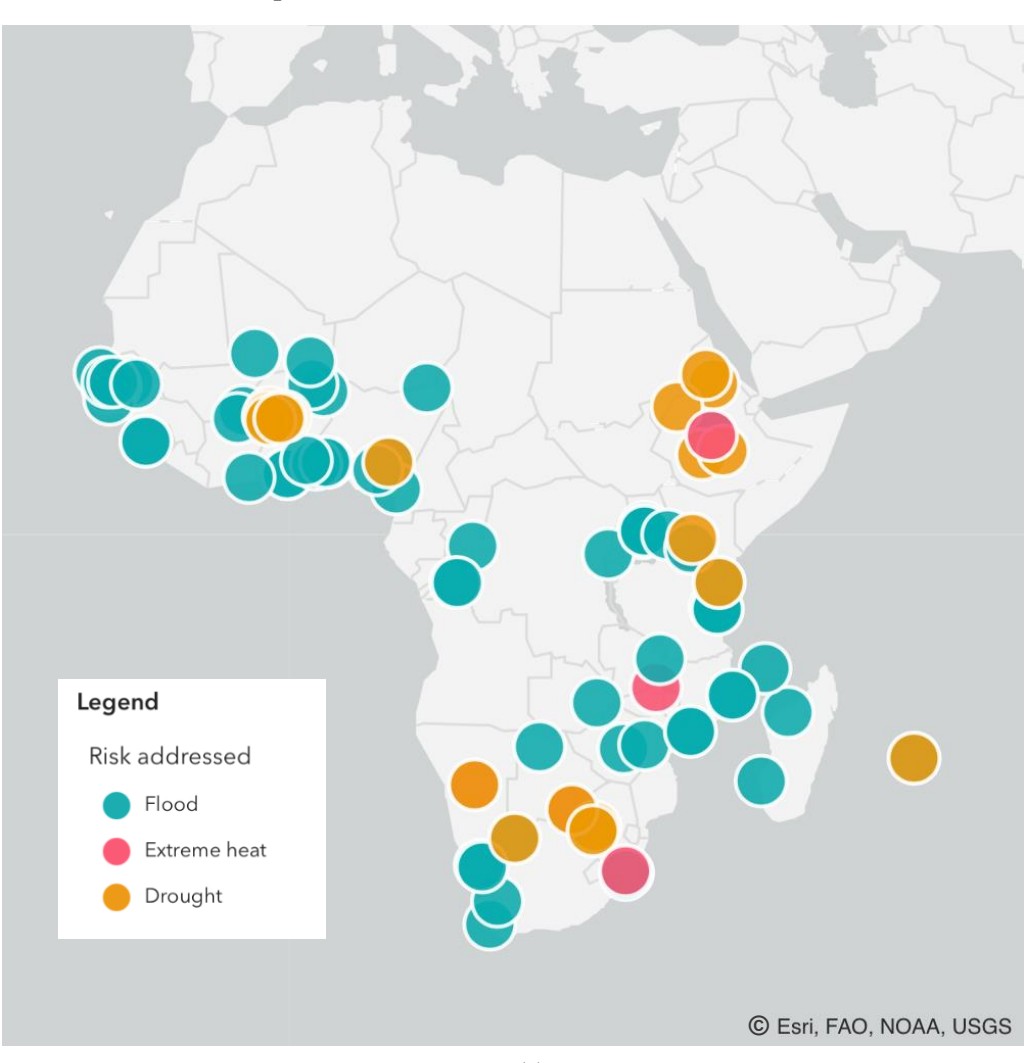

**(a)**

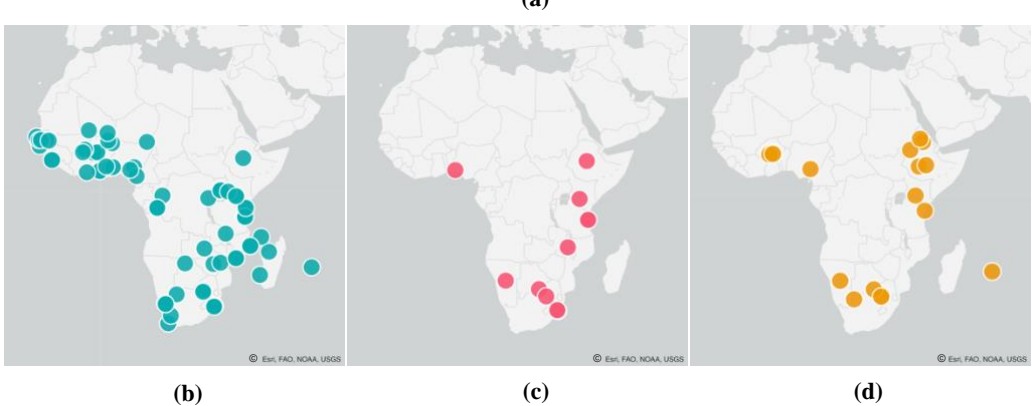

**(b)**                                    **(c)**                                    **(d)**





**Figure 5. Map indicating the locations of all the reported NBS in SSA for hydro-meteorological risks mitigation.** (a) Locations of all
risks studied; (b) Locations of papers studying floods only; (c) Locations of papers studying extreme heat only; and (d) Locations of papers
studying drought only.
For floods, the most NBS were implemented in Dar es Salaam (n=4) and Kampala (n=3), both located in East Africa. Two
NBS were implemented in Nairobi and Gazi Bay, both in Kenya in East Africa; Accra in Ghana and Lagos in Nigeria in West
Africa; Durban and Johannesburg in South Africa and Nacala and Quelimane in Mozambique in Southern Africa.
Regarding extreme heat mitigation, the most NBS (n=6) were implemented in Southern Africa. Three NBS were implemented
in East Africa, with most in Dar es Saleem (n=2). There was only one NBS in West Africa, in Lagos, Nigeria, and none were
reported in Central Africa.
For drought mitigation, the city of Johannesburg in South Africa in Southern Africa was reported to have the most NBS
implemented (n=2). Only one NBS was implemented in each of the remaining cities. However, the majority of the NBS were
clustered in West Africa (n=9), followed by East Africa (n=8) and then Southern Africa (n=3). Figure 5 presents the locations
where the NBS were implemented.

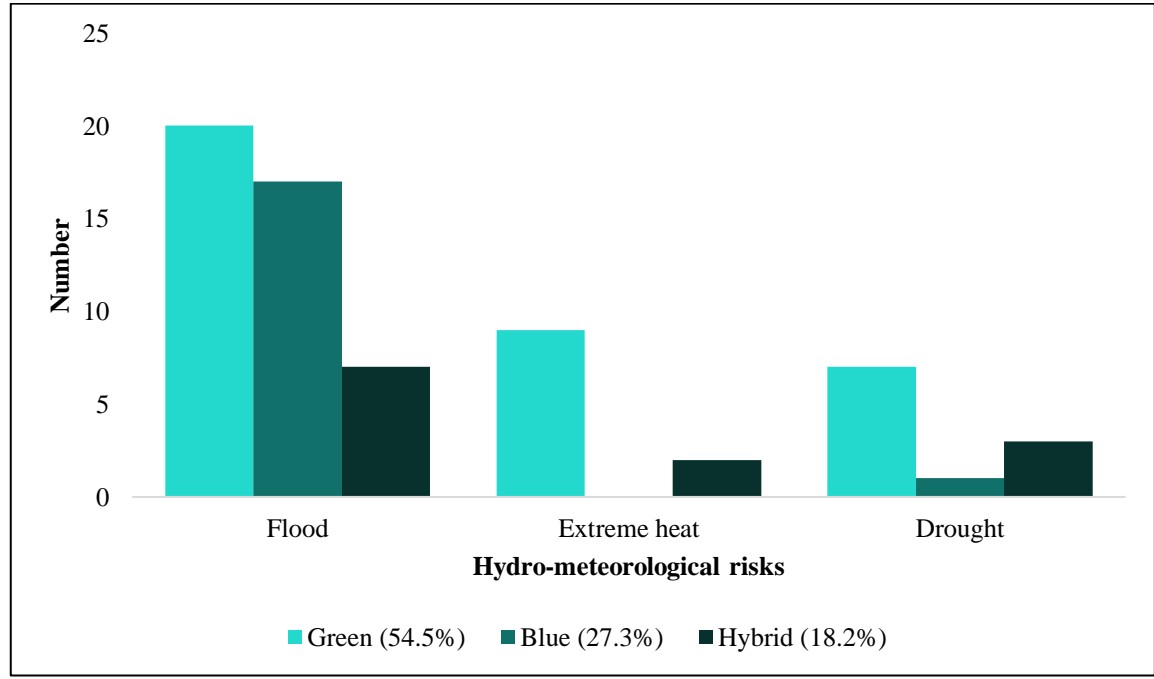

**Figure 6. The link between NBS type and risks addressed.**
The study found that green NBS (n=20) were the most widely used for flood mitigation, followed by blue NBS (n=17). Hybrid
NBS (n=7) were the least used. For extreme heat mitigation, most NBS are green (n=9), while a couple were found to be





hybrid. There were no recorded blue NBS. For drought mitigation, 7 green NBS, 3 grey measures and 1 blue NBS were
reported. Figure 6 presents the link between NBS types and the hydro-meteorological risks addressed.

### 3.3. Specific NBS types and practices in use in SSA

The study found 36 green, 18 blue and 12 hybrid NBS practices reported for mitigating floods, extreme heat and drought in
SSA. They altogether summed up to 66 different NBS practices, with 44 deployed for addressing floods, 11 for addressing
extreme heat and 11 for mitigating drought.
In terms of flood mitigation, the most reported NBS practices were mangrove restoration (n=10), wetland restoration (n=7),
urban agriculture (n=5) and marine conservation (n=5). For extreme heat mitigation, reforestation (n=10), urban forests (n=8)
green conservation (n=7), gardens (n=6) and green/open spaces (n=6) were the most reported practices. For drought, the most
common practices reported were agroforestry (n=3), conservation agriculture (n=2), integrated soil management (n=2) and
sustainable agriculture (n=2). Table 2 presents the detailed list of NBS types and practices used for hydro-meteorological risks
mitigation in SSA.
**Table 2. List of NBS types and practices used for mitigating floods, extreme heat and drought in SSA, their frequency and sources.**





| Hydro-meteorological risk addressed | NBS practice | NBS type | Frequency | Reference |
|---|---|---|---|---|
| **Flood** | Bamboo planting | Green | 1 | Mulligan et al. (2020) |
| | Constructed wetland | Blue | 1 | Mulligan et al. (2020) |
| | Coral reef restoration | Blue | 1 | Garcia (2019) |
| | Cross-cutting theme | Hybrid | 1 | Adegun et al. (2021) |
| | Floodplain conservation | Blue | 3 | Douglas (2018) Thorn et al. (2021) Turner et al. (2021) |
| | Floodplain restoration | Blue | 2 | Douglas (2018) Turner et al. (2021) |
| | Grass strips | Green | 1 | Kalantari et al. (2018) |
| | Integrated approach | Hybrid | 1 | Ajibade (2017) Kihara et al. (2020) |
| | Mangrove conservation | Green | 4 | Fischborn & Herr (2015) ICLEI (2020) Kalantari et al. (2018) Thorn et al. (2021) |
| | Mangrove restoration | Green | 10 | Fairhurst et al. (2012) Fischborn & Herr (2015) Garcia (2019) ICLEI (2020) Kalantari et al. (2018) Laros et al. (2013) Ravenholt (2021) Taylor & Oluoch (2012) UN Environment (2019b) Washbourne (2022) |
| | Marine conservation | Blue | 5 | Fairhurst et al. (2012) Fischborn & Herr (2015) Kalantari et al. (2018) Thorn et al. (2021) Washbourne (2022) |
| | Meso-scale vegetation | Green | 1 | Adegun et al. (2021) |
| | Natural fountain | Blue | 1 | Thorn et al. (2021) |
| | Natural retention ponds | Blue | 1 | Jongman et al. (2019) |
| | Parks | Green | 3 | Adegun et al. (2021) Thorn et al. (2021) Washbourne (2022) |
| | Peatland conservation | Green | 1 | Kopansky et al. (2020) |
| | Peatland restoration | Green | 1 | Kopansky et al. (2020) |
| | Permeable surfaces | Hybrid | 1 | Fairhurst et al. (2012) |
| | Pervious paving | Hybrid | 1 | Mulligan et al. (2020) |
| | Planted infiltration pits | Blue | 1 | Mulligan et al. (2020) |
| | Planted revetment | Green | 1 | Mulligan et al. (2020) |
| | Rain gardens | Green | 1 | Mulligan et al. (2020) |
| | Rainwater harvesting | Blue | 4 | Garcia (2019) Mulligan et al. (2020) Tamagnone et al. (2020) UN Environment (2019a) |
| | Rangeland rehabilitation | Green | 2 | CIFOR (2013) |





| | | | | Reid et al. (2018) |
|---|---|---|---|---|
| | Recycled and planted tires | Green | 1 | Mulligan et al. (2020) |
| | Resettlement | Blue | 3 | Douglas (2018) Kita (2017) Thorn et al. (2021) |
| | Restoration of degraded forests | Green | 1 | Global Landscapes Forum (2021) |
| | Land restoration | Green | 1 | CIFOR (2013) |
| | Revegetation of degraded slopes | Green | 1 | Doswald et al. (2021) |
| | River conservation | Blue | 1 | Laros et al. (2013) |
| | River restoration | Blue | 4 | Douglas (2018) ICLEI (2020), Thorn et al. (2021) World Bank (2020b) |
| | Sand dune | Blue | 1 | Thorn et al. (2021) |
| | Sewer connection | Hybrid | 1 | Mulligan et al. (2020) |
| | Soil remediation | Green | 1 | Mulligan et al. (2020) |
| | Springwater collection | Blue | 1 | Mulligan et al. (2020) |
| | Stone dykes | Hybrid | 1 | UN Environment (2019a) |
| | Swales | Green | 1 | Mulligan et al. (2020) |
| | Underground detention/infiltration | Hybrid | 1 | Mulligan et al. (2020) |
| | Urban agriculture | Green | 5 | Douglas (2018), Habtemariam et al. (2019) Lwasa et al. (2014) Mulligan et al. (2020) Thorn et al. (2021) |
| | Vegetated open areas | Green | 1 | Mulligan et al. (2020) |
| | Vegetative waterways | Green | 1 | Turner et al. (2021) |
| | Watershed rehabilitation | Blue | 1 | World Bank (2013) |
| | Wetland conservation | Blue | 3 | ICLEI (2010), Jongman et al. (2019) Weise et al. (2021) |
| | Wetland restoration | Blue | 7 | Benchwick (2019) CIFOR (2013) Douglas (2018) ICLEI (2010) Reid et al. (2018) UN Environment (2016) Weise et al. (2021) |
| **Extreme heat** | Gardens | Green | 6 | Adegun et al. (2021) Etshekape et al. (2018) Mugure (2020) Mulligan et al. (2020) Thorn et al. (2021) UN Environment, 2019b) |
| | Green roof | Hybrid | 1 | Adegun et al. (2021) |





| | Green conservation | Green | 7 | Etshekape et al. (2018)<br>Fischborn & Herr (2015)<br>ICLEI (2020)<br>Laros et al. (2013)<br>Mauvais (2018)<br>Washbourne (2022)<br>World Bank, 2014, 2014) |
|---|---|---|---|---|
| | Green/open spaces | Green | 6 | Habtemariam et al. (2019)<br>ICLEI (2010)<br>Laros et al. (2013)<br>Thorn et al. (2021)<br>World Bank (2020b, 2021) |
| | Green space conservation | Green | 1 | Kalantari et al. (2018) |
| | Reforestation | Green | 10 | Doswald et al. (2021)<br>Fischborn & Herr (2015)<br>GIZ (2021)<br>ICLEI (2010)<br>Ravenholt (2021)<br>UN Environment (2019b)<br>UNDP (2017)<br>World Bank (2014, 2019, 2020a) |
| | Soccer field/playground | Green | 1 | Thorn et al. (2021) |
| | Tree-planting | Green | 1 | Doswald et al. (2021) |
| | Urban forest | Green | 8 | Adegun et al. (2021)<br>Choi et al. (2021)<br>Etshekape et al. (2018)<br>Moyo et al. (2021)<br>Mulligan et al. (2020)<br>Schäffler & Swilling (2013)<br>Thorn et al. (2021)<br>Washbourne (2022) |
| | Urban greening | Green | 2 | Fairhurst et al. (2012)<br>Laros et al. (2013) |
| | Vertical greening system | Hybrid | 1 | Adegun et al. (2021) |
| **Drought** | Agroforestry | Green | 3 | Doswald et al. (2021)<br>Etshekape et al. (2018)<br>Lwasa et al. (2014) |
| | Anti-fire corridors | Hybrid | 1 | UN Environment (2019a) |
| | Climate-smart agriculture | Green | 1 | World Bank (2020a) |
| | Composting toilet | Hybrid | 1 | Mulligan et al. (2020) |
| | Conservation agriculture | Green | 2 | Kihara et al. (2020)<br>Laros et al. (2013) |
| | Organic farming | Green | 1 | UNDP (2017) |
| | Retaining walls | Hybrid | 1 | UN Environment (2019a) |
| | Integrated soil fertility management | Green | 2 | Ajibade (2017)<br>Kihara et al. (2020) |
| | Protection of water sources | Blue | 1 | Kalantari et al. (2018) |
| | Restoration of degraded land | Green | 1 | ICLEI (2010) |



| Sustainable agriculture | Green | 2 | Fischborn & Herr (2015) World Bank (2020a) |
| --- | --- | --- | --- |

*NB: Definitions of each NBS type and practice can be found in supplementary material table 4.*

### 3.3.1. Green NBS practices

Mangrove restoration (n=10) and conservation (n=4) are used for mitigating floods, especially in coastal areas and are a very popular NBS practice in SSA. Mangroves serve as natural buffers against tidal pressure and storm surges. They also provide a range of ecosystem services, including sediment stabilization, prevent saltwater intrusion into up-shore ecosystems like wetlands and provide breeding grounds for various fish, crustaceans and birds. Evidence of these benefits has been seen in Douala (Cameroon) (Lwasa et al., 2014). The potential of mangroves to capture and store carbon is being demonstrated through the restoration of mangrove areas in Cape Winelands and other locations in South Africa through the Local Action for Biodiversity project (ICLEI, 2010). Our study revealed that urban agriculture (n=5) is being used in some locations in SSA, including Accra (Ghana), Dar es Salaam (Tanzania) and Kampala (Uganda), to mitigate floods (Douglas, 2018). Urban agriculture has been found to help slow runoff by 15-20%, depending on the type of soil and amount of rainfall (Lwasa et al., 2014).

The most reported NBS practice for extreme heat mitigation was reforestation (n=10). Reforestation refers to the intentional restocking of depleted forests and woodlands. Many such efforts were found across different locations in SSA (GIZ, 2021). Urban forests are a comprehensive assemblage of trees within urban contexts. The review found urban forests to be a widely reported green NBS practice in SSA (n=8) (e.g., Adegun et al., 2021; Choi et al., 2021; Etshekape et al., 2018). Green conservation involves activities that help to protect existing trees and other forms of vegetation. Several green conservation efforts (n=7) were found in this review, with cases reported in Kinshasa (DR Congo) (Etshekape et al., 2018) and many cities in South Africa (Washbourne, 2022). Within domestic settings, studies by Adegun et al. (2021), Thorn et al. (2021), Etshekape et al. (2018) and others revealed the increasing use of gardens (n=6) for addressing many risks and providing co-benefits, including food and herbs.

There are reports of local people and urban farmers adopting agroforestry (n=3) to cope with the changing climate and associated drought events (Etshekape et al., 2018). Conservation agriculture (n=2) has also become important in Muttare, Zimbabwe, due to water scarcity (Kihara et al., 2020). Other practices identified were integrated soil fertility management (n=2) and sustainable agriculture (n=2). Integrated soil fertility management refers to a range of practices in cropping and fertilizer application, especially on small farms that seek to maximize production, while sustainable agriculture aims to bring innovation and recycling into agriculture to make it more circular. Climate-smart agriculture which seeks to adapt crop cultivation and animal rearing to the changing climate and reduce emissions from agriculture, was found in Ethiopia (n=1) (World Bank, 2020a).





### 3.3.2. Blue NBS practices

Concerning flood mitigation, wetland restoration (n=7) was the most reported blue NBS. The restoration of wetlands involves the manipulation of degraded wetlands' physical, chemical and biological characteristics to return them to their natural condition. In contrast, wetland conservation (n=3) aims to protect existing wetlands from degradation. Marine conservation encapsulates efforts to protect oceans and ecosystems in and around them from pollution and over-exploitation through planned management efforts. As revealed in the review, such efforts focused on preventing the degradation of marine ecosystems for flood protection, such as pioneering marine protected area management in Madagascar (Kalantari et al., 2018). Kalantari et al. (2018)'s study, which observed the effectiveness of rainwater harvesting technologies, showed the possibility of addressing flooding and drought concurrently in urban areas. Others have focused on the ecological restoration of rivers (n=4) under diverse pressures (e.g., Douglas, 2018; ICLEI, 2020; Thorn et al., 2021).

Regarding floodplain conservation (n=3) and restoration (n=2), which are also widely used for flood mitigation, studies by Thorn et al. (2021), Douglas (2018) and Turner et al. (2021) found many efforts across SSA. These studies found that floodplain conservation and restoration initiatives within urban settings can be challenging because of the presence of informal settlements that often make dwellings in these places and depend on the natural resource for their livelihoods. Closely related to such efforts is the resettlement of people living in the buffer zones, which also emerged in the review (n=3). In such instances, after relocation, floodplains are either conserved or restored to their natural state if degraded.

On drought mitigation, one practice, the protection of water sources, was reported in Kenya. It was aimed at enhancing water availability by providing more watering points in national parks and community areas (Kalantari et al., 2018). No blue practices were found for extreme heat mitigation.

### 3.3.3. Hybrid NBS practices

Each of the 12 hybrid NBS practices identified was reported only once. They ranged from quite traditional practices such as the use of stone dykes and retaining walls in Comoros for flood mitigation (UN Environment, 2019a) and composting toilets in Kenya to more widely accepted practices like green roofs and vertical greening systems in Nigeria (Adegun et al., 2021) for extreme heat and flood mitigation, and pervious paving in Kenya for flood mitigation (Mulligan et al., 2020).

### 3.4. Ecosystem services and economic benefits provided

Ecosystem services are either provisioning, regulatory or cultural (TEEB, 2010). Intrinsically, NBS used for mitigating hydro-meteorological risks provide regulatory ecosystem services, whether flood control, reversing the impact of extreme heat or addressing drought. However, the study sought to explore if other ecosystem services were provided beyond the studied hazard mitigation services (Fig. 7).




Twenty-four different ecosystem services made up of five different provisioning services (20.8 %), 15 regulatory services
(62.5%) and four cultural services (16.7 %) were identified. In all, 88.9% (40 papers) reported at least one type of ecosystem
service, while 11.1% (5 papers) reported none. 13.3% (6 papers) reported on only one type of ecosystem service, 46.7% (21
papers) reported on two types of ecosystem services, and 28.9% (13 papers) reported on all three types of ecosystem services.
**Figure 7. Ecosystem services provided by NBS initiatives beyond studied hazards mitigation.**





### 3.4.1. Provisioning services

Provisioning services provide direct benefits to urban residents, such as water, food, fuel and herbs. The review found that poor households in many informal settlements in cities depend directly on these provisioning services for their subsistence and livelihoods. In coastal areas and floodplains, fisheries and aquaculture were found to be more popular (e.g., Douglas, 2018; Ibe & Sherman, 2002; Turner et al., 2021), while food crops, fuel and herbs were found to be more common inland (Kihara et al., 2020; Lwasa et al., 2014; Schäffler & Swilling, 2013). For instance, in Obudu, Nigeria, the community is reported to have planted over 4,000 threatened *afang* vine and bush mango seedlings as part of reforestation efforts, which provide edible non-timber forest products such as nuts and fruits (UNDP, 2017).

### 3.4.2. Regulatory services

The predominant regulatory service reported was carbon sequestration (n=26). In Durban, the Buffelsdraai Landfill Site Community Reforestation Project was conceived before the 2010 FIFA World Cup and aimed to see over 500 thousand indigenous trees planted. This restoration project was anticipated to help "absorb event-related greenhouse gas emissions while enhancing the capacity of people and biodiversity to adapt to the inevitable effects of climate change" (Douwes et al., 2015, p.6). The Great Green Wall project, which is roughly 15% underway, is expected to sequester 250 million tons of $CO_2$ by 2030 (Turner et al., 2021). Some studies acknowledged the importance of urban green areas for providing shade, reducing fire risk, increasing soil biodiversity and serving as windbreaks, among others (e.g., Etshekape et al., 2018; Kihara et al., 2020; Moyo et al., 2021). Other authors studied how urban greens help control erosion (n=17) both along the coasts (e.g., Fischborn & Herr, 2015; Ibe & Sherman, 2002; ICLEI, 2020) and inland (e.g., Adegun et al., 2021; Kalantari et al., 2018). Furthermore, restoration programmes are helping to maintain habitats and populations (n=9), especially in monitoring the loss of threatened species, ecosystems and critical habitats (Doswald et al., 2021). Weise et al. (2021) found that wetland conservation and restoration programmes are helping to protect thousands of bird and fish species across Botswana and Burkina Faso.

### 3.4.3. Cultural services

The cultural services provided were recreation (n=13), aesthetic value (n=4), education and research (n=2) and cultural heritage (n=1). In South Africa, the reforestation efforts under the Buffelsdraai Landfill Site Community Reforestation Project and the construction of the Buffelsdraai Reforestation Hub, which was an educational centre, provided recreation for residents and tourists. A review in Nigeria found similar benefits for green spaces (Adegun et al., 2021). Also, studies by Habtemariam et al. (2019) and Thorn et al. (2021) found that different NBS had aesthetic values that helped improve the image of cities. Papers describing various NBS projects in Ethiopia (ICLEI, 2020), Botswana, Zimbabwe, Tanzania and others found the same (Laros et al., 2013). In the Succulent Karoo in South Africa, the restoration of wetlands for flood mitigation also led to the creation of sites of value in the wetland areas for education and research purposes (Reid et al., 2018). A similar outcome was found in Lagos in Nigeria, where the Lekki Urban Forest and Animal Sanctuary helped to address extreme heat (Mauvais, 2018).



### 3.4.4. Livelihood and income generation

Ecosystem services provide a range of benefits, including social benefits such as improved human health and wellbeing, social cohesion and reduced crime, and economic benefits such as job creation and income generation. Thirty-four of the included papers (75.6%) reported on livelihood generation. Notably, most livelihood generation opportunities created were green jobs in disciplines like horticulture, forestry and market gardening. Cases from Kenya show that NBS for hydro-meteorological risks mitigation can create employment in the designing, planning, implementation and post-project phases (Mulligan et al., 2020). According to Doswald et al. (2021), restoration programmes can promote small businesses and increase household incomes.

For NBS with an international implementation scale, the Gazi Mangrove Restoration project in Kenya is reported to employ dozens of people and attract over 300 eco-tourists each month (Taylor & Oluoch, 2012). To address gender inequalities, the jobs created through the project were reserved for women only.

With regional NBS, the Great Green Wall across the width of Africa created 350 thousand green jobs as of 2018 after its inception in 2007, mainly through land restoration activities, employment of rangers and nature guards and the production and sale of non-timber forest products. About $89.9 million was generated in revenue through these activities over the same period. The green job potential of the project is expected to reach 10 million by 2030 (UNCCD, 2020).

In the context of national NBS, Moyo et al. (2021) report that the Buffelsdraai Landfill Site Community Reforestation Project in South Africa created employment during the planting period between 2008 and 2016. Specifically, 50 full-time, 16 part-time and 389 temporary jobs were created. Over 600 tree-pruners were also reported to be supplying seedlings to the project in exchange for vouchers to buy food, bicycles, pay for school fees and vehicle driving lessons, especially during the planting phase. In addition, there is an opportunity to upscale these livelihood benefits by utilizing invasive species such as *Chromolaena odorata, Melia azedarach* and *Eucalyptus*, which invaded the project site. In Uganda, a wetlands restoration project advanced by the United Nations Development Programme is expected to help improve the lives of over 500 thousand people, including providing them with livelihood options (Benchwick, 2019). A tree-planting programme in Freetown, Sierra Leone, also helped to create 550 short-term jobs focusing on women, youth and marginalized groups (Ravenholt, 2021).

At the community level, the rangeland rehabilitation and wetland restoration initiative in the Succulent Karoo of South Africa accentuates the potential of NBS for green job creation. It is reported that "937 jobs were created through two public works programmes funded by the DEA Expanded Public Works Programme Natural Resource Management Programme and building on CSA project activities (De Villiers 2013) – 611 jobs under the 'Working for wetlands' programme activities (implemented by South African National Parks), and a further 326 jobs under the 'Working for water' programme implemented by CSA between 2014 and 2017" (Reid et al., 2018, p. 12-13). These green jobs were mainly in restoration activities.



## 4. Discussion

### 4.1. Extent of reported NBS for hydro-meteorological risk mitigation uptake in SSA

After conducting this systematic review, we find that SSA is critically understudied in the area of NBS for hydro-meteorological risks mitigation. Du Toit et al. (2018) found that only 38% of cities in SSA had any research carried out on them on green infrastructure and ecosystem services. Choi et al. (2021)'s review of green infrastructure found that only 1% of the included papers were from Africa. Nevertheless, there may be more NBS initiatives in SSA, except that they are unreported or were not captured within the search terms used in this study. Such unreported NBS most likely draw on local knowledge and are community-based, which makes documenting them challenging, a challenge linked to an ineffective data management culture in SSA (Malgwi et al., 2020; Manteaw et al., 2022). It is also likely that those locations in which NBS are reported in the scientific literature are locations where research funds have been made available for their investigation. What is more, there may be other activities that could qualify as NBS but are not so described. For example, African farmers have been using NBS-like practices like agroforestry, stone bunds, grass strips and sustainable land use through techniques like observing fallow periods for generations without calling them NBS (Keesstra et al., 2018). As such, it is unclear where the fine line should be drawn between age-old traditional practices and NBS or whether they should be considered NBS at all. Adopting the co-created citizen science approach, which brings lay people and experts together for knowledge co-creation (Gill et al., 2021), could help to incorporate such traditional practices, which are effective, into NBS and promote inclusivity and sustainability. The present study, therefore, affirms assertions that literature on NBS and hydro-meteorological risks mitigation in SSA is scant, but this may be due in part to a lack of documentation and use of different terminologies.

The study finds that most papers were from South Africa, Kenya, Nigeria and Tanzania. This could be because these countries are among the biggest economies in SSA—South Africa and Nigeria, in particular, are the two biggest economies in SSA (Kamer, 2022)—and are somewhat leaders in their respective sub-regions. The four countries have also been forerunners in incorporating concepts like green infrastructure in urban planning, especially South Africa (e.g., Frantzeskaki et al., 2019; Russo et al., 2017; Venter et al., 2020). Furthermore, they boast of some of the best educational and research institutions, which places them in a good position to advance research on urbanization, climate change and concepts like NBS and ecosystem services.

Most reported NBS were implemented on the national scale. This is likely because major climate funds like Global Environment Facility and Climate Fund are more easily accessible to national governments than non-profit and community-based organizations. Notwithstanding, local-scale NBS are the second most common. Such initiatives are often grassroots-driven, which enables local people to maximize benefits. Many challenges often constrain local governance in SSA—decentralization mechanisms may be ineffective, local-level capacity may be weak and financial resources may be limited (Hjerpe et al., 2014). For many SSA countries, development and climate adaptation often occur only when driven from the grassroots by non-state actors or when local institutions are robust enough to lead or coordinate initiatives (Mubaya &





Mafongoya, 2017). The Local Action for Biodiversity project advanced by ICLEI (which focused on improving the capacity
of local governments and political actors, including mayors, on biodiversity and ecosystems) presents a good case study of
how national, even regional and international projects can support local communities to develop more sustainably.
International and regional NBS also promote knowledge-sharing, which is essential, especially in applying a novel concept
like NBS and in the context of the shared climate crisis that confronts all regions of the world.

### 6 4.2. Relationship between location of NBS and location of risks

Although they are not located in the areas the IPCC predict will receive the harshest climate impacts in SSA, Somalia, South
Sudan and populations along the coast of Mozambique are identified as the most vulnerable to hydro-meteorological risks due
to poor household and community resilience, high population densities and weak governance systems (Busby et al., 2014). In
this review, only Mozambique, among these most vulnerable countries, reported NBS.
Based on the total deaths recorded from climate-related disasters, Somalia, Mozambique, and Nigeria have been the most
affected (CRED, 2019). However, only Nigeria, third on the list, emerges among the most studied countries in this review.
**Table 3. Top countries impacted by weather-related disaster deaths in SSA against top sources of papers by country in this review.**

| Country | Total deaths | Country | No. of papers |
|---|---|---|---|
| Somalia | 20,739 | South Africa | 8 |
| Mozambique | 3,777 | Kenya | 8 |
| Nigeria | 1,696 | Tanzania | 6 |
| Madagascar | 1,644 | Nigeria | 6 |
| Ethiopia | 1,639 | Uganda | 4 |
| Kenya | 1,572 | SSA | 4 |
| Sierra Leone | 1,289 | Senegal | 4 |
| DR Congo | 1,072 | Ethiopia | 4 |
| Malawi | 985 | Madagascar | 3 |
| | | Ghana | 3 |
| | | DR Congo | 3 |
| | | Burkina Faso | 3 |

*Source: CRED (2019) and results from this review.*
The factors behind very few papers from the most at-risk countries could be attributed to political instability. Somalia, in
particular, is third globally and first in SSA on the Global Fragile States Index (Nasri et al., 2021). South Sudan, fourth globally
and second in SSA on the Global Fragile States Index, is also a relatively new country. Other reasons may be lack of capacity
for developing winning proposals for accessing climate funds and dwindling climate finance globally. The exclusion of papers
published in Portuguese because they are not as widely spoken as English and French could have also led to the low



identification of papers in countries like Mozambique, Sao Tome and Angola. The present study, therefore, asserts that reported
NBS for hydro-meteorological risks mitigation in SSA are located in areas where risks are, but not where they are most severe.
In SSA, blue NBS have been the most used when addressing floods, while green NBS are more popular for extreme heat and
drought mitigation. However, in Europe, hybrid practices are the most popular when addressing floods, while green NBS are
more prevalent when responding to heatwaves and droughts. Blue NBS are the least used (Sahani et al., 2019). NBS
implementation often demands land, such as river restoration, which is often unavailable due to urbanization (Pugliese et al.,
2022). In Europe, 90% of floodplains have been ecologically degraded (Entwistle et al., 2019), and the sections of urban areas
vulnerable to floods increased by 1,000% between 1870 and 2016 (Paprotny et al., 2018). These factors have hampered the
uptake of blue and green NBS, which is why practitioners have had to settle for hybrid NBS practices. In SSA, the rapid rate
of urbanization often makes it challenging for city officials to keep up with urban environmental change, which is characterized
by greens depletion and environmental degradation (Cobbinah et al., 2019). Even so, the proliferation of blue and green NBS
implies that decision-makers can structure urbanization using lessons from the Global North to avoid counterproductive
practices and develop climate-resiliently. In particular, lessons can be drawn from NBS like the Isar River restoration in
Germany (Pugliese et al., 2022) and the implementation of constructed wetlands, bio-swales, permeable pavements and other
NBS in the sponge city concept in China (Li & Zhang, 2022), both for flood mitigation; as well as ambitious greening efforts
across Europe (Pauleit et al., 2019), Singapore and Hong Kong to improve thermal comfort (Aflaki et al., 2017).

### 4.3.    Specific NBS types and practices in use in SSA

Out of 66 NBS practices identified, most were implemented for flood mitigation. Earlier studies had found that 64% of hazard
events in Africa from 2000 to 2019 were flood-related (CRED, 2019). However, many identified NBS were reported to address
more than one risk (Fig. 3). This demonstrates the multi-functionality of NBS and highlights their relevance for SSA in the
efforts to address the multiplicity of challenges in the sub-region within the context of limited climate adaptation funds.
Comparatively, Sahani et al. (2019) found 205 NBS used for addressing floods, heatwaves and drought in Europe. In a review
in the German Alps, Zingraff-Hamed et al. (2021) also found 156 NBS used to address floods and landslides. While NBS is
gradually becoming popular in SSA, it has not seen the level of wide uptake in the Global North, despite being the most
vulnerable to hydro-meteorological risks.
Regarding flood mitigation, the most reported NBS were mangrove restoration and wetland restoration. For extreme heat
mitigation, reforestation, urban forests and green conservation measures were the most reported NBS. In Europe, NBS like
river and floodplain restoration (Zingraff-Hamed et al., 2021) and natural water retention measures (Hartmann et al., 2019) are
more widely used for flood mitigation, while different green infrastructure types are used for heatwave mitigation (Pauleit et
al., 2019). In this review, the most commonly reported NBS for drought mitigation were agroforestry, conservation agriculture,
integrated soil management and sustainable agriculture. Thus, there may be many similarities between NBS practices used in





SSA and Europe. However, food production appears to be a critical necessity for many SSA locals, even in the uptake of NBS
for hydro-meteorological risks mitigation. Indeed, the agricultural sector is one of the most sorely affected by climate change
in SSA (Stringer & Dougill, 2013), and it is predicted that yields could drop to up to 50% by 2100 (FAO, 2009). This could
explain why communities often lend more support to NBS projects that provide provisioning ecosystem services like fruits
from tree crops (Etshekape et al., 2018).
NBS practices that are not common in SSA but are more widely used in the Global North were identified in SSA. These include
green roofs, vertical greening, constructed wetlands and soil remediation. Green roofs are building rooftops where plants are
grown in extensive or intensive ways. The review found the increasing use of green roofs in many locations in Nigeria (Adegun
et al., 2021). Vertical greening systems are plants grown along the vertical axis of buildings, either on the façade or in the
interior. Studies in Nigeria found the practice to improve thermal conditions and provide edible and medicinal plants
(Akinwolemiwa et al., 2018; Oluwafeyikemi & Julie, 2015). Soil remediation is the process through which soils are returned
to their original form of ecological stability before being disturbed. In Kenya, this method was used to help address floods
through reduced runoff and improve access to co-benefits such as agricultural lands (Mulligan et al., 2020). These buttresse
the assertation that there may be many similarities between NBS practices used in Europe and those used in SSA.
### 4.4.   Ecosystem services and economic benefits provided
SSA's most critical challenges include food and water insecurity, poverty, unemployment and climate change (World
Economic Forum, 2019). Fifty percent of people in SSA live in urban areas (Kelsall et al., 2021), and over 43% of this urban
population live below the poverty line (Du Toit et al., 2018). Most of these people live in informal spheres and lack access to
decent and affordable housing, food and water and other necessities of life (Güneralp et al., 2017). Provisioning ecosystem
services such as food, water and fuel are therefore necessary. This explains the popularity of NBS, which are closely related
to food provision—agriculture already employs most of the labour force—such as agroforestry and climate-smart agriculture.
Also, the urban poor are the most vulnerable to climate change impacts, and that NBS can provide livelihood options is
welcomed by locals. For decision-makers, the evidence that NBS can promote climate action through carbon sequestration,
mitigate heat and beautify cities, among others, are important benefits and drivers of adoption (Lupp, Huang, et al., 2021).
Aside from delivering hazard mitigation services, NBS could help address some of SSA's developmental challenges
concurrently.
Cultural ecosystem services provide non-material benefits such as recreation, education and intellectual appreciation, physical
and mental benefits, aesthetic significance, spiritual and symbolic appreciation and enjoyment (Roux et al., 2020). The majority
of the papers did not report on cultural ecosystem services. This study then adds to a long list of studies highlighting how
cultural ecosystem services are little researched (e.g., Jones et al., 2022; Milcu et al., 2013).  The lack of data in this sense
makes it challenging to demonstrate the full spectrum of the benefits and dis-benefits of NBS. It reiterates calls by earlier



authors to scientists to produce ecosystem services assessment frameworks, especially for cultural ecosystem services, to
improve reporting (Christie et al., 2019; Schäffler & Swilling, 2013).
Most of the papers included in the review reported that NBS created livelihood opportunities. Creating livelihood opportunities,
particularly green jobs, which are more sustainable, is important for a youthful region like SSA, where 60% of the population
is 25 years or younger (Mo Ibrahim Foundation, 2019). This is also relevant in addressing crime and insecurity, which is often
rife among the 50% and over people who reside in informal spheres in urban SSA due to lack of economic opportunities. Plus,
improving life standards may reduce the destruction of natural habitats and enhance natural restoration. Despite this, livelihood
generation needs to be studied in detail, especially in river conservation and restoration projects, because in some instances,
NBS led to the loss of livelihoods of local people. These have often occurred where risk responses have required the
resettlement of populations; an NBS found to be used in SSA in this study. While its consideration as an NBS on its own may
be contestable, Douglas (2018) indicates that relocation of informal settlements within riparian zones is a significant part of
conservation and restoration initiatives in many locations in SSA, such as in Nairobi, Kenya. When such informal settlers were
offered compensation and alternative livelihood options and relocated, they preferred to move back to these riparian areas,
even if they were at risk of being impacted by floods because their livelihoods were tied to these places. When river corridors
have also been improved, it increased the value of such lands, becoming more attractive to developers and displacing the
original informal settlers. This mirrors concerns with conventional engineering solutions like wastewater treatment plants,
raises critical social justice concerns and could lead to a critique of the NBS concept.

## 18    5.   Conclusions

This review presented an overview of NBS for hydro-meteorological risks mitigation in urban areas of SSA, considering the
extent of uptake, the location of NBS in relation to where risks are, the specific NBS types and practices in use and the benefits
derived by way of ecosystem services and livelihood opportunities created.
From the analysis of 45 papers, we found at least one reported NBS in 71% of urban areas of SSA countries. However, this
does not tell the whole story, as more than half of the NBS were based in only four countries. Hence, we conclude that first,
reported uptake of NBS for hydro-meteorological risks in SSA is low. However, there could be more ongoing NBS, especially
at the community level, that are unreported. Second, NBS are implemented where risks are but not where they are most severe.
Third, there are many similarities between NBS practices used in SSA and Europe. Even practices like green roofs, vertical
greening and constructed wetlands, which are more used in the Global North, are emerging in the sub-region. Fourth, food
provision is, in most cases, a key objective of NBS in locales, even in hazard mitigation, with NBS like agroforestry and
gardens used quite significantly. Fifth, the proliferation of blue and green NBS in SSA indicates that the sub-region can advance
urban development in a greener way and avoid repeating past mistakes in the Global North that led to the depletion and
dwindling of green and blue spaces. Sixth, designing NBS inclusively can help to address challenges that confront localities



more head-on since many SSA countries have difficulties with the over-centralization of governance and ineffective local government systems. Seventh, NBS could help address some of the major developmental challenges that confront SSA, including water and food insecurity, unemployment and poverty, aside from climate change and the associated hydro-meteorological risks. Eighth, if not inclusively designed, planned and implemented, NBS can affect livelihoods, as seen in the case of resettlement as part of efforts to conserve or restore floodplains and other vital ecosystems. This may raise crucial social justice concerns about the NBS concept. Ninth, the concept of NBS needs to be further debated to clarify its scope, including its principles and use within different regional contexts. Apart from the consideration of conservation efforts as NBS, this review also showed that the use of traditional methods like grass strips—which fit the definition of NBS—hundreds of years ago in SSA raises the question of whether such age-old traditional practices should be considered NBS.

From a policy perspective, we recommended that the concept of NBS is incorporated into urban planning in SSA to help address socio-ecological challenges associated with urban sprawl, such as greenspace depletion, water-related ecosystems degradation and pollution while helping to build resilience against hydro-meteorological risks. Adopting a co-created citizen science approach which will help increase knowledge on NBS and incorporate local knowledge into NBS interventions, is also recommended. Given that food production, which is threatened by climate change, is a key objective for locals even in the roll-out of NBS for hydro-meteorological risks mitigation, we recommend decision-makers prioritize NBS that promote urban and peri-urban agriculture. Furthermore, we propose that knowledge exchange opportunities on NBS be explored between SSA countries where the concept is still emerging, Europe and other regions where there has been widespread uptake.

For future studies, we recommend that more quantitative research produce or update risk and vulnerability maps, assess the effectiveness of individual NBS and study the multifunctionality of NBS in terms of ecosystem services and social and economic benefits. Research studying conventional engineering solutions and NBS comparatively, using, for instance, experimental set-ups, modelling or expert interview approaches, are also encouraged. Understanding the ecosystem disservices of NBS, such as increased abundance of diseases causing insects like mosquitoes that carry malaria and increased harassment in green corridors, can also be advanced to understand the pros and cons of NBS fully.

**Data availability**

No datasets were used in this article.

**Author contributions**

KBE, AZ-H and MAR conceived the research, its design and analysis. AZ-H and SP led in the structuring and organization of the paper. KBE led in the data collection and analysis. AZ-H, MAR and LCS contributed to the analysis. KBE led in authoring the manuscript. LCS contributed to authoring the discussion section of the manuscript. SP reviewed and streamlined the draft manuscript.



**Conflict of interest**
We wish to confirm that this publication has no known conflicts of interest. We confirm that the manuscript has been read and
approved by all named authors and that there are no other persons who satisfied the criteria for authorship but are not listed.
**Acknowledgements**
Gratitude goes to Susanne Raum (PhD) for her helpful guidance in undertaking systematic reviews during the
conceptualization of this paper and Titouan Dubo for his assistance in the database search.
**Funding**
The authors greatly appreciate the Andrea von Braun Stiftung, which provided funding for this PhD project.

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
