# Peer review of "Review article: Potential of Nature-Based Solutions to Mitigate Hydro-Meteorological Risks in Sub-Saharan Africa"

_EGUsphere, 2022_

## Author Response (AR1)

**POINT-BY-POINT RESPONSE TO REFEREES**

**Referee #1**

Authors: We would like to thank Referee #1 for their precious time in reviewing our manuscript. The insightful comments have helped to make key improvements to the manuscript. We carefully considered the comments and tried our best to address every one of them. Below, responses are provided to the specific comments and suggestions that were made.

Referee #1: Page 2 lines 17-26 - Before even the term NBS was born, other approaches for hydrometeorological risk reduction were adopted both without or in combination with conventional engineering measures. These were the so called soft engineering measures, which were more environmentally friendly. The authors should put more efforts in contextualizing the concept of NBS with prior similar terminologies, such as Ecological Engineering or Soil and Water Bioengineering, which have many aims and applications in common with NBS and recent studies have compared terms and defintions of these with NBS.

Authors: This comment is well noted. We have included a table (1) on concepts related to NBS such as low-impact development, green infrastructure, ecosystem-based adaptation, sustainable urban drainage systems and ecosystem-based disaster risk reduction in addition to Ecological Engineering and Soil and Water Bioengineering as suggested. This table also explains how these related concepts are linked to NBS and which regions they have been mostly used in. This is available on page 2 line 28-page 3 lines 1.

Referee #1: Page 3 line 1- typo, it should be "locals" instead of "locales"

Authors: Locales as used here is referring to a place, but we have replaced it with "local communities".

Referee #1: Pae 3 line 8 - please remove the bold font for "protect, conserve"

Authors: This is well noted and has been addressed accordingly.

Referee #1: Page 4 line 19 - other sister projects of the H2020 projects mentioned should be also nominates (e.g. Operandum, Reconect etc)

Authors: This is well noted and other recent EU H2020 projects like OPERANDUM, Reconnect, ThinkNature and CLEARING HOUSE have been acknowledged as well. We have also indicated that as of September 2021; 32 projects had been funded under the initiative across 59 countries.

Referee #1: Page 5 line 1 - maybe I was distracted but I think that the first time I see that we are talking about "urban" NBS is here. Please mention that you are talking about urban areas also in the introduction. And also, if urban NBS is the focus, why is not present in the selection criteria shown in Figure 1?

Authors: The first sentence of the Introduction set the context of the study to be urban areas and on page 3 line 28, it was reiterated. We have, however, elaborated the reason for focusing on urban reasons, being engines of growth, having high population densities and the fact that SSA is undergoing rapid urbanization in the revised manuscript. This is available on page 4 lines 29-32 and page 5 lines 1-3.

Referee #1: Table 1: I think that here again Ecological Engineering or Soil and Water bioengineering, as well as urban forestry should be mentioned.

Authors: Thank you for this suggestion. Table 1 contains the terms that were used, in different combinations, for the literature search process. Urban forests were included but ecological engineering and soil and water bioengineering were not. This is because the selection of our search terms was informed by the examination of other papers on NBS, including the three below:

Ruangpan, L., Vojinovic, Z., Di Sabatino, S., Leo, L. S., Capobianco, V., Oen, A. M., ... & Lopez-Gunn, E. (2020). Nature-based solutions for hydro-meteorological risk reduction: A state-of-the-art review of the research area. *Natural Hazards and Earth System Sciences*, 20(1), 243-270.

Du Toit, M. J., Cilliers, S. S., Dallimer, M., Goddard, M., Guenat, S., & Cornelius, S. F. (2018). Urban green infrastructure and ecosystem services in sub-Saharan Africa. *Landscape and Urban Planning*, 180, 249-261.

Thorn, J. P. R., Aleu, R. B., Wijesinghe, A., Mdongwe, M., Marchant, R. A., & Shackleton, S. (2021). Mainstreaming nature-based solutions for climate resilient infrastructure in peri-urban sub-Saharan Africa. *Landscape and Urban Planning*, 216, 104235.

We did not include ecological engineering and soil and water bioengineering in our search terms since we did not find them in these papers.

We also examined the literature for NBS typologies (e.g., Somarakis, G., Stagakis, S., Chrysoulakis, N., Mesimäki, M., & Lehvävirta, S. (2019). ThinkNature nature-based solutions handbook) but did not find ecological engineering and soil and water bioengineering, which is why we did not include these terms.

This has been clarified in the revised manuscript as well and is available on page 8 lines 2-4.

Referee #1: Paragraph 2.5. Study Limitation - The auhtors have checked also possible projects implemented by humanitarian associations or Engineers without borders, who might have worked in these countries in the past with works related to Soil and Water Bioengineering,

Authors: We appreciate the suggestion. We searched the websites of 12 key institutions and 11 NBS databases where the majority of NBS projects in SSA are reported. Even so, we acknowledge that our search may not be exhaustive which is why we note the dependence on reported NBS as a study limitation.

Referee #1: Figure 2 - this figure would be more complete,  if you can add also the location of the cities where you foind NBS are implemented, being this a review on urban NBS

Authors: Thank you for this suggestion. We had included the NBS locations in Figure 2 initially but later decided to leave them out as they were already indicated in Figure 5. We have included the specific city locations into Figure 2 with small yellow markers in the revised manuscript.

Referee #1: Page 10 lines 6-9 - you found 45 papers but how many NBS? this is not specified.

Authors: We indicated the number of NBS we identified, which were 66 in all. These were detailed on Pages 16 lines 4-12 and in Table 2 (Table 3 in revised manuscript).

Referee #1: Figure 5 - Maybe try to make the markers smaller.

Authors: This is well noted and has been addressed accordingly.

Referee #1: Table 2 - you have classified the NBS in three types: blue, geen and hybrid appraches. Can you please explain how you have grouped them and on what is based your classification?

Authors: The classification of the NBS into green, blue and hybrid was explained in the Introduction on Page 3, lines 18-27 (page 4 lines 19-28 in the revised manuscript). Also in the revised manuscript, this has been included in bracket in the title of Table 3.

Referee #1: Figure 7 Please specify the numbers after each term, where they come from?

Authors: These numbers were generated automatically along the sankey diagram (in https://sankeymatic.com) based on the number of reported ecosystem services but are not relevant for understanding the figure. Many thanks for this comment. The numbers have been masked.

Referee #1: Page24 lines 20-21- can you please explain how these species created opportunities to upscale livelihood benefits?

Authors: This comment has been addressed accordingly. Many thanks for pointing it out. It has been explained that these species are useful, for instance, for medicinal purposes, in the revised manuscript. The Chromolaena odorata can be used to treat skin ailments, the Melia azederach for controlling diabetes and gastrointestinal disorders and the Eucalyptus as an antioxidants and insect repellent.

Referee #1: Table 3 - maybe you can optimize this table and keep only one column with the country, so it's easier to make comparisons.

Authors: This comment is well noted and only the first two columns have been kept.

Referee #1: Page 29 lines 16-17 - This last sentence is not clear to me. Can you please explain this concept further?

Authors: While several examples of conventional engineering solutions, as used in this study, were given in the manuscript, we acknowledge that they were not explicitly defined. As such, a definition has been provided in the revised manuscript (page 2 lines 17-19) that conventional engineering solutions refer to grey infrastructure solutions that often have no or very minimal room for nature and are rarely multi-functional. Thanks again.

**Referee#2**

Authors: We would like to appreciate Referee #2 for their time and effort in reviewing our manuscript and providing insightful feedbacks. These insightful comments have helped to improve the manuscript. We carefully considered each comment and tried our best to address them. Details are presented below.

Referee #2: Page 3 line 15 – here biodiversity is mentioned, but why not say something more specific about flora and fauna?

Authors: This comment is well appreciated, however, since we were only noting the principles of NBS in urban areas in this section, we thought a generalized term was more appropriate.

Referee #2: Page 3 line 23-24 – I suggest to mark the number of categories as follows: (i) green, (ii) blue, (iii) hybrid….

Authors: This is well noted, and the numbering has been included in the revised manuscript.

Referee #2: Page 4 line 7 – Go a bit deeper on Ecosystem Services (ES) theme.

Authors: This is well noted and has been addressed accordingly. We have included a definition of ecosystem services. We have also provided examples of the three ecosystem service types such as food and fuel, erosion control and heat mitigation and recreation and aesthetic value respectively for provisioning, regulatory and cultural ecosystem services (page 5 lines 13-16).

Referee #2: Page 4 line 23-24 – I suggest to integrate Evans et al. (2022) "Ecosystem service delivery by urban agriculture and green infrastructure – a systematic review" as citation.

Authors: This citation has been included in the revised manuscript.

Referee #2: Page 5 line 1-6 – please, check if paper answers all questions.

Authors: The paper does answer all four research questions. To help make the reading clearer, we have repeated the research questions (objectives) in the Conclusions and re-organized the study conclusions in the order of appearance of the study objectives.

Referee #2: Page 10 line 6 – How many SSA States are there in which there is not even a studio? Are they particularly concentrated in a specific area? If yes, why?

Authors: The number of SSA countries that did not report any NBS at all were 14 since there are 48 SSA countries in all according to the World Bank. In terms of the

sub-regional concentration of the NBS, these were explained in page 10 lines 6-9. Of the reported NBS, 30 were in West Africa, 18 in Southern Africa, 30 in East Africa and 6 in Central Africa. We note, however, that most of the papers were from only four countries (South Africa, Kenya, Nigeria and Tanzania). We explain that these countries are among the biggest economies in SSA, have been forerunners in incorporating concepts like green infrastructure and NBS in urban planning and have the best research institutions in the sub-region. This perhaps, is the reason why they produced many studies on the concept.

Referee #2: Page 12 Figure 4 – Is part of this caption not transferable into the text?

Authors: Indeed, the explanation of the different scales of NBS were initially part of the text but we thought that it was better to include it in the figure caption so that it is easier to interpret the figure and still maintain so.

Referee #2: Page 22 Figure 7 – Define ESS acronym.

Authors: Thanks for this comment. It has been addressed accordingly.

Referee #2: Page 23 Line 9 – What about some regulatory services on water cycle?

Authors: This comment is well acknowledged however, we were taking stock of what ecosystem services have been reported and as such, were limited to only that. Unfortunately, none of the included papers reported on how land cover can regulate water flow.

Referee #2: Page 27 Line 12 – please deepen the part about Global North, also included in the conclusions.

Authors: The comment is well appreciated and the discussions in the Global North has been increased accordingly in the revised manuscript (page 28 lines 12-17.

Referee #2: Supplementary Table 4 – typo, it should be "open spaces" instead of "opens paces" to define parks as NBS practice.

Authors: The comment is well acknowledged, and the correction has been made in the revised supplementary materials document.

Referee #2: I also suggest a whole review of the paper to improve communication effectiveness and, for the future, to work on on research activities that can overcome the study limitations set out in sub-chapter 2.5.

Authors: These comments are well acknowledged. We have thoroughly reviewed the manuscript accordingly and suggested future studies that could overcome the identified study limitations in the revised manuscript. This includes proposing for

future studies to use surveys and/or interviews as well to reduce full dependence on only reported NBS to be able to assess the success or failure of projects to document lessons by collecting empirical data. This is available on page 31 lines 30-32 and page 32 lines 1-5. Many thanks once again.